# RIS-Assisted Hybrid Beamforming and Connected User Vehicle Localization for Millimeter Wave MIMO Systems

**DOI:** 10.3390/s23073713

**Published:** 2023-04-03

**Authors:** Md. Abdul Latif Sarker, Woosung Son, Dong Seog Han

**Affiliations:** 1Center for ICT & Automotive Convergence, Kyungpook National University, Daegu 41566, Republic of Korea; 2Graduate School of Electronic and Electrical Engineering, Kyungpook National University, Daegu 41566, Republic of Korea; 3School of Electronics Engineering, Kyungpook National University, Daegu 41566, Republic of Korea

**Keywords:** millimeter-wave MIMO, RIS, vehicle-to-vehicle communications, connected autonomous vehicle, spectral efficiency, location error

## Abstract

A reconfigurable intelligent surface (RIS) is a type of metasurface that can dynamically control the reflection and transmission of electromagnetic waves, such as radio waves, by changing its physical properties. Recently, RISs have played an important role in intelligently reshaping wireless propagation environments to improve the received signal gain as well as spectral efficiency performance. In this paper, we consider a millimeter wave (mmWave) vehicle-to-vehicle (V2V) multiple-input multiple-output (MIMO) system in which, an RIS is deployed to aid downlink V2V data transmission. In particular, the line-of-sight path of the mmWave system is affected by blockages, resulting in higher signaling overhead. Thus, the system performance may suffer due to interruptions caused by static or mobile blockers, such as buildings, trees, vehicles, and pedestrians. In this paper, we propose an RIS-assisted hybrid beamforming scheme for blockage-aware mmWave V2V MIMO systems to increase communication service coverage. First, we propose a conjugate gradient and location-based hybrid beamforming (CG-LHB) algorithm to solve the user sub-rate maximization problem. We then propose a double-step iterative algorithm that utilizes an error covariance matrix splitting method to minimize the effect of location error on the passive beamforming. The proposed algorithms perform quite well when the channel uncertainty is smaller than 10%. Finally, the simulation results validate the proposed CG-LHB algorithm in terms of the RIS-assisted equivalent channel for mmWave V2V MIMO communications.

## 1. Introduction

Millimeter wave (mmWave) communication is a promising technology that plays a major role in fifth-generation (5G) wireless communications owing to its wide bandwidth [1], sufficient spectral-efficiency [2,3,4,5] and high quality of service (QoS) [6,7,8,9,10]. In highly directive wireless communications, beamforming is a signal processing tool used to steer, shape, and control multiple antenna directions efficiently, which minimizes transmission losses [11,12,13,14,15,16]. The use of fully digital beamforming has been widely adopted in traditional communication systems, but it requires a radio frequency (RF) chain per antenna element, which increases power consumption, hardware complexity, and costs. As a solution to this problem, hybrid beamforming has been proposed as an alternative approach for mmWave massive multiple-input and multiple-output (MIMO) systems [17,18]. To further develop traditional hybrid beamforming, the proposed conjugate gradient-based hybrid beamforming algorithm can deal with the user sub-rate maximization problem and offer the benefit of significantly improving spectral efficiency.

The authors developed an adaptive algorithm in [2] to estimate the mmWave channel parameters that exploit the poor scattering nature of this channel. In [2], focus on both the single and multi-path channels to validate the adaptive channel estimation algorithm. A generalized hybrid architecture was proposed with a small number of RF chains and a finite number of ADC bits and the authors derived achievable rates with channel inversion and singular value decomposition-based transmission methods. A content-centric communications scenario was studied and a beamforming-based rate-splitting method applied to maximize the sum rates of all users in [11]. In addition, joint beamforming and power allocation was demonstrated for a non-orthogonal multiple access (NOMA)-based satellite-terrestrial integrated network in [12]. In [13], a receiver prototype was presented that enables reconfiguration between analog and digital beamforming modes to better deal with different operating scenarios. The authors designed a true-time-delay analog and digital beamformer in [13]. Elsewhere, a network was proposed featuring a multi-antenna array base station (BS) and a reconfigurable intelligent surface (RIS) to deliver information to information users and power to energy users [14]. Conjugate and zero-forcing beamforming was proposed to develop a path-following algorithm to solve the user throughput maximization problem subject to a realistic constraint on the quality-of-energy-service in terms of the harvested energy thresholds in [14]. In [15], the authors investigated a secure transmission design for an intelligent reflecting surface (IRS)-assisted unmanned aerial vehicles network and solved active and passive beamforming and trajectory optimizations. In addition, a neural networks-based beam codebook was designed in [16]. There are three hybrid beamforming architectures investigated in the literature: fully connected, fixed subarray and dynamic subarray. In [17], an iterative hybrid beamforming design algorithm was proposed with a fully connected architecture to maximize the weighted sum-rate performance of mmWave MIMO systems. The authors considered an RIS-aided mmWave-NOMA downlink system with a hybrid beamforming structure in [18]. They developed an alternating optimization algorithm to solve the non-convex optimization issue. Finally, geometric mean decomposition-based beamforming for RIS-assisted mmWave hybrid MIMO systems was proposed in [19]. The authors designed phase shifters for RIS by maximizing the array gain for the line-of-sight (LoS) channel.

Recently, mmWave communication has included a vehicle-to-everything (V2X) paradigm to facilitate connected autonomous driving, video entertainment, vehicle platooning, and dynamic map update [20]. All of these services require high-speed connections to exchange safety information (e.g., vehicle location, direction and speed) among adjacent vehicles since recent V2X standards such as cellular V2X in [21], do not have advanced QoS and cannot provide consistent high-speed data transmission. Owing to the sparsity of the mmWave channel [1], the MIMO system typically requires an LoS connection, which maintains the received power level. Indeed, the LoS channel between the transmitter and receiver may be blocked by static or moving objects, such as buildings, trees, vehicles, and humans [22]. To satisfy the service quality of V2X communication under LoS blockage, RISs have also been represented as large intelligent surfaces (LISs), which have been investigated as an emerging technology in [6,7,23]. An RIS controller with beamforming is typically offered with high-accuracy indoor or outdoor radio localization [24]. In mmWave MIMO communication, the RIS can easily establish robust connections subject to blockage or mobility awareness. RISs are mainly used as antenna arrays for increasing coverage area and reducing energy consumption [25]. Moreover, RISs are applied as reflective relays [26]. The functionalities of RIS controllers increase the quality of the received signal when the adjustable passive elements can individually steer an incident electromagnetic wave in any direction.

Based on the RIS element arrays, beamforming and steering concepts have been investigated in [23,24,25,27]. Due to the notable benefits of RIS technology, outdoor coverage was analyzed using RISs in [25] and IRS-aided beamforming concepts discussed in [27]. A channel estimation algorithm was proposed in [28,29], an RIS-aided manifold optimization algorithm was developed in [30], and an RIS-aided real-time autonomous vehicle model was investigated in [31]. Elsewhere, researchers proposed ultra-reliable low-latency communication in a factory automation (FA) scenario to deploy an IRS to create an alternative transmission link, which can enhance transmission reliability [32]. The performance of single and multiple RIS-assisted systems has also been considered without a direct path between the transmitter and the receiver in indoor and outdoor propagation environments [33]. In addition, researchers proposed a geometry-based stochastic channel model that supports the movements of transceivers and clusters, and the evolution of clusters was considered in the space domain, where a reflecting coefficients design was based on the minimum path loss [34]. Elsewhere, researchers provided a comprehensive overview of state-of-the-art research on emerging and promising RIS/IRS-aided wireless systems, with an emphasis on signal processing techniques for solving various radio localization, transmission design, and channel estimation issues [35]. A tutorial overview of single and multi-IRS-aided wireless networks has also been provided, with an emphasis on addressing the new and more challenging issues in IRS reflection optimization and channel acquisition design, in [36]. In [37], a multi-IRS-aided system was studied in which, the IRS and base station (BS) are managed by a central processing unit to coordinate data transmission and maximize the weighted sum rate of all cell-edge users by jointly optimizing the BS’s transmit beamforming and each IRS’s phase shifts, subject to the BS transmit power budget. The authors solved the non-convex and unit modulus constraint optimization using an efficient iterative power allocation algorithm. The achievable secrecy performance of mmWave MIMO systems was studied using RISs in [38], where the authors assumed a smart environment in which an RIS is placed between the source and the legitimate user to enhance the main link. The authors, motivated by the aforementioned observations, studied beamforming optimization for an IRS-aided multi-antenna communication system by incorporating signal distortions caused by hardware impairments [39]. The authors optimized both source’s transmit active and passive beamforming to maximize the SNR received at the destination. Elsewhere, the application of an active refracting RIS-enabled transmitter for a secure internet of things network was investigated to enhance secure communication in the considered network and develop an alternating optimization algorithm to optimize the sum secrecy rate by jointly designing the power allocation, transmit beamforming, and the phase shifts of an RIS in [40].

Due to the high beam training costs and the effect of user location errors in vehicular MIMO communication systems, we propose an RIS-assisted location-based hybrid beamforming algorithm as well as a double-step iterative algorithm for minimizing the effect of user vehicle location error, which has not yet been investigated in the literature. The contributions of this study are listed as follows:Due to blockage awareness, we first develop an RIS-assisted V2V MIMO channel model and demonstrate the geometrical relation between the RIS controller and user vehicle position. By using the distance between the transmitting and receiving vehicle in relation to the distance of an RIS controller, we estimate the path amplitude and phase.We then design an RIS-assisted low-dimensional equivalent effective channel. In response to a sub-design problem of an original beamformer, we propose a conjugate gradient and location-based hybrid beamforming algorithm. We apply a Karush-Kuhn-Tucker condition and a Lagrangian method to solve an original problem into a sub-optimal problem for developing the conventional hybrid beamforming algorithm, which reduces the user sub-rate maximization problem. The proposed algorithm attains significant spectral efficiency performance.We next consider a covariance splitting method to reform the error covariance matrix. Since the channel is geometrically contained in location information, we propose a double-step iterative algorithm that minimizes the effect of location error in user vehicles. Finally, the simulation results demonstrate the superiority of the proposed algorithms over their counterparts.

The rest of the paper is organized as follows. The signal and channel model and the geometrical relation between the RIS controller and connected autonomous vehicle position are discussed in Section 2. The proposed hybrid beamforming and user vehicle localization is presented in Section 3. Simulation results are provided in Section 4. The conclusions are offered in Section 5.

## 2. Signal and Channel Model

### 2.1. Signal Model

Consider an access point that serves a point-to-point (P2P) vehicular MIMO network using mmWave technology, as shown in Figure 1, where a transmit vehicle is operating at mmWave frequency bands with $N_{t}$ transmit antennas. The user vehicle is equipped with $N_{u}$ receive antennas. Let the transmit and user vehicles be autonomous, which are assumed to be fully connected with different modules and sensors as the power module is used as a battery to supply power to the other module. The perception module is recognized for driving environments and detected objects using sensors such as light detection and ranging (LiDAR), radars, cameras, a global positioning system (GPS), and inertial measurement unit (IMU), and other sensors as shown in Figure 2. A WiFi-based mmWave transceiver is also intelligently connected to the transmit and receive vehicles in Figure 2. We consider an IEEE 802.11ad Wi-Fi chips with the number of 8 antennas to communicate at the 60 GHz frequency band.

A blockage scenario is considered in the system model where the direct link between the transmit and user vehicles is blocked owing to the critical propagation conditions. Hence, the RIS is deployed in the transmission model to assist and expand the service coverage of the P2P vehicular MIMO communication. The received signal vector $r\in C^{N_{u}\times 1}$ can be modeled as
(1)$$r=H_{e}x+z,$$
where $x\in C^{N_{t}\times 1}$ is the transmitted signal vector, which satisfies $E[xx^{H}]=I$, $H_{e}$ is the $N_{u}\times N_{t}$ effective channel from the transmitting vehicle to the user vehicle, $z∼\mathrm{CN}(0,\sigma_{z}^{2}I)$ is the $N_{u}\times 1$ additive white Gaussian noise vector and $\sigma_{z}^{2}$ denotes the noise variance.

After applying a hybrid precoding, x is given by
(2)$$x=F_{A}F_{D}s,$$
where $F_{A}\in C^{N_{t}\times N_{RF}}$ is the analog radio frequency (RF) beamforming matrix, $F_{D}\in C^{N_{RF}\times N_{s}}$ is the digital precoding matrix, $\left|\right|F_{A}F_{D}{\left|\right|}_{F}^{2}\leq P$, *P* is the transmit power, $s\in C^{N_{s}\times 1}$ is the transmitted symbol vector that satisfies $E[ss^{H}]=I$, $N_{s}\leq \min\left(N_{RF}^{t},N_{RF}^{u}\right)$ when $N_{t}>>N_{RF}^{t}$, $N_{u}>>N_{RF}^{u}$, $N_{RF}^{t}$ denotes the number of RF chains for the transmit vehicle side, and $N_{RF}^{u}$ denotes the number of RF chains for the user intelligent vehicle side.

Setting (2) in (1) and the received signal vector $\hat{s}$ is given after combining
(3)$$\hat{s}={\tilde{H}}_{e}s+\tilde{z},$$
where ${\tilde{H}}_{e}=W_{D}^{H}W_{A}^{H}H_{e}F_{A}F_{D}$ is the $N_{s}\times N_{s}$ effective equivalent channel, $\tilde{z}=W_{D}^{H}W_{A}^{H}z$ is the $N_{s}\times 1$ equivalent noise vector, $W_{D}\in C^{N_{RF}\times N_{s}}$ is the baseband digital combiner, and $W_{A}\in C^{N_{u}\times N_{RF}}$ is the analog combiner which follows the similar constraints as $F_{A}$. If the transmitted signal follows a Gaussian distribution over the RIS-assisted mmWave V2V MIMO channel, then using (3), the achievable spectral efficiency is given by
(4)$$R=\log_{2}\det\left(I+E^{-1}{\tilde{H}}_{e}{\tilde{H}}_{e}^{H}\right),$$
where $E=\sigma_{z}^{2}W_{D}^{H}W_{A}^{H}W_{A}W_{D}$ is the noise covariance matrix after combining and the operator ${\left(\cdot \right)}^{H}$ denotes the Hermitian.

### 2.2. Channel Model

Let $H_{e}=G\Theta C$ where G is an $N_{u}\times N_{R}$ channel between RIS to the user vehicle as
(5)$$G=\gamma_{g}\sum_{l=1}^{L_{g}}{\tilde{\beta }}_{l}a_{u}\left(\phi_{u}^{l}\right)a_{R}^{H}\left(\theta_{t}^{l},\nu_{t}^{l}\right),$$
where ${\tilde{\beta }}_{l}$ denotes the complex channel gain at *l*-th path of G channel, $\phi_{u}^{l}$ represents the angle of arrival associated with the user vehicle, $\gamma_{g}=\sqrt{N_{u}N_{R}/L_{g}}$, and $\theta_{t}^{l}$ and $\nu_{t}^{l}$ denotes the azimuth and elevation angles of arrival and departure at *l*-th path, respectively, the normalized array response vector for the case of the uniform linear array is given by [2,3,4,5].
(6)$$a_{u}\left(\phi_{u}^{l}\right)=\frac{1}{\sqrt{N_{t}}}{\left[1,\;e^{\frac{j2\pi d\sin\left(\phi_{u}^{l}\right)}{\lambda }},\ldots ,e^{\frac{j2\pi \left(N_{t}-1\right)d\sin\left(\phi_{u}^{l}\right)}{\lambda }}\right]}^{T},$$
where the wavelength $\lambda =c/f_{c}$, *c* is the speed of light, $f_{c}$ is the carrier frequency, and d=λ/2 is the antenna spacing. The normalized array response vector for uniform planner array is given by [28]
(7)$$\begin{array}{cc} a_{R}\left(\theta_{t}^{l},\nu_{t}^{l}\right) & =\frac{1}{\sqrt{N_{t}}}[1,\;e^{\frac{j2\pi d}{\lambda }\left(\left(n_{1}-1\right)\cos\left(\nu_{t}^{l}\right)\sin\left(\theta_{t}^{l}\right)+\left(n_{2}-1\right)\sin\left(\nu_{t}^{l}\right)\right)},\\ \; & \ldots e^{\frac{j2\pi d}{\lambda }\left\{\left(N_{R,y}-1\right)\cos\left(\nu_{t}^{l}\right)\sin\left(\theta_{t}^{l}\right)+\left(N_{R,z}-1\right)\sin\left(\nu_{t}^{l}\right)\right\}}{]}^{T}, \end{array}$$
where $N_{R}=N_{R,y}\times N_{R,z}$, and C denotes the $N_{R}\times N_{t}$ channel between the transmit vehicle and the RIS controller, that is
(8)$$C=\gamma_{c}\sum_{l=1}^{L_{c}}{\tilde{\gamma }}_{l}a_{R}\left(\theta_{r}^{l},\nu_{r}^{l}\right)a_{t}^{H}\left(\phi_{t}^{l}\right),$$
where ${\tilde{\gamma }}_{l}$ denotes the complex channel gain at *l*-th path of C channel, $\gamma_{c}=\sqrt{N_{t}N_{R}/L_{c}}$, $a_{R}\left(\theta_{r}^{l},\nu_{r}^{l}\right)$ and $a_{t}^{H}\left(\phi_{t}^{l}\right)$ are designed in the same manner of G channel and the $N_{R}\times N_{R}$ RIS element response matrix Θ is given by
(9)$$\Theta =\frac{1}{\sqrt{N_{R}}}D,$$
where D is the diagonal phase matrix as [25],
(10)$$D=\mathrm{diag}\{\vartheta_{1},\ldots ,\vartheta_{N_{R}}\},$$
where $\vartheta ={\left[\vartheta_{1},\ldots ,\vartheta_{N_{R}}\right]}^{T}$, $\vartheta_{n}=e^{j\varphi_{n}}$, $n=1,2,\cdots ,N_{R}$ and $\varphi_{n}$ denotes the phase-shift which is given by
(11)$$\varphi =\left[\begin{array}{ccc} \varphi_{1,1} & \cdots & \varphi_{1,N_{R}}\\ \vdots & \ddots & \vdots \\ \varphi_{N_{R},1} & \cdots & \varphi_{N_{R},N_{R}} \end{array}\right].$$

Thus, the effective channel $H_{e}$ can be written as
(12)$$H_{e}=\sum_{m=1}^{L_{c}}\sum_{l=1}^{L_{g}}\beta_{l}\gamma_{m}\mu_{l,m}a_{u}\left(\phi_{R}^{l}\right)a_{t}^{H}\left(\theta_{t}^{l}\right),$$
where $\mu_{l,m}=a_{R}^{H}\left(\theta_{t}^{l},\nu_{t}^{l}\right)\Theta a_{R}\left(\theta_{r}^{m},\nu_{r}^{m}\right)$ denotes the passive beamforming gain, $\beta_{l}=\gamma_{g}{\tilde{\beta }}_{l}$, $\gamma_{l}=\gamma_{c}{\tilde{\gamma }}_{l}$, and $\mu_{l,m}$ satisfy $|\gamma_{1}\left|\geq \right|\gamma_{2}\left|,\ldots ,\geq \right|\gamma_{L_{c}}|$ and $|\beta_{1}\left|\geq \right|\beta_{2}\left|,\ldots ,\geq \right|\beta_{L_{g}}|$.

### 2.3. Geometrical Relation between the RIS Controller and the Connected Vehicle Position

Let $b=(b_{x},b_{y},b_{z})$, $r=(r_{x},r_{y},r_{z})$, and $u=(u_{x},u_{y},u_{z})$ are the position of the transmitting vehicle, the RIS controller and the user vehicle, respectively in the Cartesian coordinate as shown in Figure 1, and g is the vector of G channel. The g is given by
(13)$$g=\mathrm{vec}\left(G\right)=a_{R}\left(\theta_{t}^{az},\nu_{t}^{el}\right),$$
where $\theta^{az}$ denotes the azimuth of the angle of departure (AoD) and $\nu^{el}$ denotes the elevation of AoD at the RIS side. The geometric relation between the RIS controller and the user vehicle position is measured by
(14)$$\theta_{t}^{az}=\arcsin\left\{\frac{u_{x}-r_{x}}{\sqrt{{\left(u_{x}-r_{x}\right)}^{2}+{\left(u_{y}-r_{y}\right)}^{2}}}\right\},$$
and
(15)$$\nu_{t}^{el}=\arcsin\left\{\frac{u_{z}-r_{z}}{||u-{r||}_{2}}\right\}.$$

Similarly, we can make a positional relationship between the transmitting vehicle and the RIS controller as
(16)$$\theta_{r}^{az}=\arcsin\left\{\frac{r_{x}-b_{x}}{\sqrt{{\left(r_{x}-b_{x}\right)}^{2}+{\left(r_{y}-b_{y}\right)}^{2}}}\right\},$$
and
(17)$$\nu_{r}^{el}=\arcsin\left\{\frac{r_{z}-b_{z}}{||r-{b||}_{2}}\right\}.$$

Using (13), the passive beamforming gain $\mu_{l,m}$ is computed as
(18)$$\mu =g^{H}\Theta g={\left[g⊙g^{*}\right]}^{H}\vartheta ,$$
where Θ=diag{ϑ} and the operator ${\left(\cdot \right)}^{*}$ denotes the conjugate, and ⊙ denotes the element-wise product. The optimal phase-shift $\vartheta^{opt}$ is given by
(19)$$\vartheta^{opt}=\underset{\vartheta }{arg max}\left|{\left[g⊙g^{*}\right]}^{H}\vartheta \right|$$

## 3. Proposed CG-LHB Algorithm and User Vehicle Localization

### 3.1. Proposed CG-LHB Algorithm

Let $N_{s}=N_{RF}$, $F=F_{A}F_{D}\in C^{N_{t}\times N_{s}}$, and $W=W_{A}W_{D}\in C^{N_{u}\times N_{s}}$. The objective function is to maximize the rate of the user vehicle while satisfying the maximum transmission power constraint. Using (4) and (12), the sub-rate maximization problem is written as
(20)$$\begin{array}{cc} \; & \max_{F,W,\vartheta }f\left(F,W,\vartheta \right)=\sum_{i}^{N_{s}}log_{2}\left(1+\frac{P}{N_{s}\sigma_{z}^{2}}\tilde{H_{e}}{\tilde{H_{e}}}^{H}\right){\;,s.t.\;||F||}_{F}^{2}\leq P,\\ \; & {\left|F\left(i,j\right)\right|}^{2}={\left|W\left(i,j\right)\right|}^{2}=1,\forall_{i,j},\;\mathrm{and}\;{\left|\vartheta_{n}\right|}^{2}=1,\forall_{m=1,\ldots ,N_{R}}, \end{array}$$
where $i=1,2,\cdots ,N_{s}$. The sub-rate maximization problem (20) is a sub-problem with respect to the digital beamformer $F_{D}$. To maximize the sub-rate, $F_{D}$ should be optimized to minimize $\left|\right|F_{A}f_{D,i}{\left|\right|}^{2}$ based on the following multi-object optimization problem (20) as
(21)$$\begin{array}{cc} \; & \min_{f_{D,i}}\left|\right|F_{A}f_{D,i}{\left|\right|}^{2},\;i=1,\ldots ,N_{s},\\ \; & \;s.t.\;H_{e}F_{A}f_{D,i}=I, \end{array}$$
where $F_{A}$, $W_{A}$ and ϑ are fixed. We see from (21), the $N_{s}$ column of $F_{D}$ can be optimized separately according to
(22)$$\begin{array}{cc} \; & \min_{f_{D,i}}\left|\right|F_{A}f_{D,i}{\left|\right|}^{2},\;i=1,\ldots ,N_{s},\\ \; & \;s.t.\;H_{e}F_{A}f_{D,i}=\varsigma_{i}, \end{array}$$
where $\varsigma_{i}$ denotes an all-zero vector except for the *i*-th entry being unity. Using the stream-specific error expressions in (22), we apply Karush-Kuhn-Tucker (KKT) conditions and the Lagrangian method for the convex problem as
(23)$$\frac{\partial f(f_{D,i},\kappa )}{\partial f_{D,i}}={\left(f_{D,i}^{H}Af_{D,i}\right)}^{T}+{\left(\kappa^{T}H_{e}F_{A}f_{D,i}\right)}^{T},$$
where $A=F_{A}^{H}F_{A}$ denotes the equivalent $N_{s}\times N_{s}$ analog beamforming matrix, κ is the Lagrangian multiplier and the corresponding Lagrangian is
(24)$$f\left(f_{D,i},\kappa \right)=f_{D,i}^{H}Af_{D,i}+\kappa^{T}\left(H_{e}F_{A}f_{D,i}-\varsigma_{i}\right).$$

If $\partial ff_{D,i},\kappa )/\partial f_{D,i}=0$, then $f_{D,i}$ and κ is given by
(25)$$f_{D,i}=-A^{-1}F_{A}^{H}H_{e}^{H}\kappa ,$$
and
(26)$$\kappa =-{\left[H_{e}F_{A}A^{-1}F_{A}^{H}H_{e}^{H}\right]}^{-1}\varsigma_{i}.$$

Thus, combining (25) and (26), the digital beamforming vector $f_{D,i}$ is given by
(27)$$f_{D,i}=A^{-1}F_{A}^{H}H_{e}^{H}{\left[H_{e}F_{A}A^{-1}F_{A}^{H}H_{e}^{H}\right]}^{-1}\varsigma_{i}.$$

Since $F_{A}$ is a constant modulus in (27), we optimize the phase of the analog beamformer for transforming a complex problem into an unconstrained optimization problem. Let the beamforming design problem can be written by
(28)$$\begin{array}{cc} \left(F_{A}^{opt},F_{D}^{opt}\right) & =\underset{F_{A},F_{D}}{arg min}\left|\right|F^{opt}-F_{A}F_{D}{\left|\right|}_{F},\;s.t.\;F_{A}\in F_{A}, \end{array}$$
where $F_{A}$ is the set of the analog beamformer, $F^{opt}=V_{1}$, which is measured by using a singular value decomposition method of $H_{e}=U\Gamma V^{H}$ in [41]. Consider $F_{A}$ into $F_{A,m}$ as
(29)$$F_{A,m}=\frac{1}{\sqrt{N_{t}}}e^{j\Omega_{m}},$$
where $\Omega_{m}$ denotes the $N_{t}\times N_{s}$ phase shift matrix, which phases need to be quantized into the nearest point in $\left(2\pi m/2^{b},m=0,1,\ldots ,2^{b}-1\right)$, if the phase shifters are of infinite or finite *b*-bit resolution. Hence, the phase optimization problem can be expressed as
(30)$$\Omega_{m}^{opt}=\underset{\Omega_{m}}{arg min}f\left(\Omega_{m}\right).$$
where $\Omega_{m}$ is converged using a conjugate gradient method, which optimizes along the direction $\Delta \Omega_{m}$
(31)$$\Omega_{m+1}=\Omega_{m}+\Delta \Omega_{m},$$
where $\Delta \Omega_{m}=\eta_{m}\Xi_{m}$ and $\Delta \Omega_{m}$ satisfy $\Delta \Omega_{m}\leq \tau$, $\eta_{m}$ is determined by minimizing $f(\Omega_{m+1})$, $\Xi_{m}$ is given by
(32)$$\Xi_{m}=-\nabla f\left(\Omega_{m}\right)+\xi \Xi_{m-1},$$
and the ξ is given by
(33)$$\xi =\frac{\nabla f{\left(\Omega_{m}\right)}^{T}\nabla f\left(\Omega_{m}\right)}{\nabla f{\left(\Omega_{m-1}\right)}^{T}\nabla f\left(\Omega_{m-1}\right)}.$$

Using (29) to (33), we summarize the proposed CG-LHB algorithm in Algorithm 1.
**Algorithm 1** Proposed CG-LHB Algorithm1:**Input parameters:**H, *b*, τ, *J*2:**Output:**$F_{A}^{opt},F_{D}^{opt}$.3:**Begin:** Set m=0, and $\Omega_{0}\in R^{N_{t}\times N_{s}}$4:If m=0,1,…,∞,$\;\Xi_{m}=-\nabla f\left(\Omega_{m}\right)$  else compute (31)–(33).5:If $\eta_{m}$ is determined by minimizing $f(\Omega_{m+1})$ in (31), then$\;\Omega_{s}\left(m+1\right)$←$f(\Omega_{m+1})$;$\;f_{s}\left(m+1\right)=f\left(\Omega_{m+1}\right)$.6:$\Delta f=f_{s}\left(m+1\right)-f_{s}\left(m\right)$;$\;\Delta \Omega =\Omega_{s}\left(m+1\right)-\Omega_{s}\left(m\right)$,  If $\Delta f\leq \tau_{1}$;  stop  If $\Delta \Omega^{T}\Delta \Omega \leq \tau_{2}$;  stop  If m+1=J;   stop  If $\nabla f{\left(\Omega_{m+1}\right)}^{T}\nabla f\left(\Omega_{m+1}\right)\leq \tau_{3}$;  converged  *m*←m+1  Go To Step 4.7:if b<∞ then8:$\Omega^{opt}=f\left(\Omega^{opt}\right)$9:Compute $F_{A}^{opt}$10:Compute $F_{D}^{opt}$.

### 3.2. User Vehicle Localization

Let RIS to user channel covariance $\Sigma_{i}=E\left[g_{i}g_{i}^{H}\right]$, where $i=1,\ldots ,N_{s}\}$ and the channel vector g=vec(G) is geometrically generated for the different positions as shown in Section 2.3. Thus, the user vehicle localization problem can be computed as a hypothesis testing problem as follows:(34)$$H_{i}:g∼\mathrm{CN}\left(0,\Sigma_{i}\right).$$

Hence, the optimal likelihood (ML)-based localization problem is given by
(35)$$\hat{i}=\underset{1\leq i\leq N_{s}}{arg max}p_{i}\left(g\right),$$
where $p_{i}\left(g\right)$ is the probability density function under G
(36)$$p_{i}\left(g\right)=\frac{1}{\pi^{N_{u},N_{t}}\left|\Sigma_{i}\left(\alpha \right)\right|}e^{-g^{H}\Sigma_{i}^{-1}\left(\alpha \right)g}.$$

We can now evaluate the error-probability performance under $H_{i}$, which is given by
(37)$$P_{e}=\frac{1}{N_{s}}\sum_{i=1}^{N_{s}}P_{e,i},\;P_{e,i}=P\left(\hat{i}\neq i|H_{i}\right).$$
where $P_{e,i}$ is the conditional error probability under $H_{i}$.

Due to user mobility and high location errors, (36) contains a high channel state information (CSI) error. To overcome this issue, we analyze the location information generated by GPS. Consider the location of the user vehicle from the GPS represented by
(38)$$u=\hat{u}+\Delta u,$$
where $u={\left[u_{x},u_{y},u_{z}\right]}^{T}$ denotes the actual location vector of the user vehicle, $\hat{u}={\left[{\hat{u}}_{x},{\hat{u}}_{y},{\hat{u}}_{z}\right]}^{T}$ is the ego GPS location vector, and $\Delta u={\left[\Delta u_{x},\Delta u_{y},\Delta u_{z}\right]}^{T}$ is the localization error vector and Δu satisfy ${\left||\Delta u|\right|}_{2}\leq \epsilon$. Based on (13) and (38), we obtain the actual channel vector g that satisfy
(39)$$g\left(u\right)=\hat{g}\left(u\right)+\Delta g\left(u\right),$$
where Δg∼N(0,Σ) and $\Sigma ={\left[\sigma_{g,x}^{2},\sigma_{g,y}^{2},\sigma_{g,z}^{2}\right]}^{T}$ is approximated as diagonal with $\sigma_{g,x}^{2}$, $\sigma_{g,y}^{2}$ and $\sigma_{g,z}^{2}$ being the channel variances along the *x*, *y*, and *z* directions, respectively. Hence, by using (39), with the location error bound (36), the CSI error minimization problem can be formulated as
(40)$$\begin{array}{cc} \; & \min_{\Delta g}f\left(\Theta ,\Delta g\right)=\left|\right|\hat{g}\left(u\right)+{\Delta g\left(u\right)||}^{2},\;\;s.t.\;||\Theta_{i,i}\left|\right|=1,\forall_{i=1,\ldots ,N_{R}},\\ \; & |a_{R}{\left(\theta_{R}^{l},\nu_{R}^{l}\right)}_{i}|=1,\forall_{i=1,\ldots ,N_{R}}{,\mathrm{and}\;||\Delta g||}_{2}\leq \epsilon . \end{array}$$

It is noted that (40) still suffers high localization errors. This issue can be addressed using a double-step iteration method. Let the error covariance matrix Σ possesses and split two different covariance matrices $\Sigma =\Sigma_{1}+\Sigma_{2}$ in (34), where $\Sigma_{1}=F^{*}\Lambda_{1}F$, $\Sigma_{2}={\hat{F}}^{*}\Lambda_{2}\hat{F}$, $\Lambda_{1}$ and $\Lambda_{2}$ are the diagonal matrices holding the eigenvalues of $\Sigma_{1}$ and $\Sigma_{2}$, respectively. If the matrices $\Sigma_{1}$ and $\Sigma_{2}$ are positive definite and α is a positive constant, then the iterative covariance matrix Σ(α) is given using by [42]
(41)$$\Sigma \left(\alpha \right)=M_{2}^{-1}N_{2}M_{1}^{-1}N_{1},$$
where α denotes the positive constant, $M_{2}=\left(\alpha I+\Sigma_{2}\right)$, $N_{2}=\left(\alpha I-\Sigma_{1}\right)$, $M_{1}=\left(\alpha I+\Sigma_{1}\right)$ and $N_{1}=\left(\alpha I-\Sigma_{2}\right)$. Based on (34), we demonstrate the proposed double-step iterative algorithm for the user vehicle localization in Algorithm 2. The double-step at each $\Sigma_{1}\Sigma_{2}\Sigma_{1}\Sigma_{2}$ iterations is necessary to obtain an exact solution for the $\alpha I+\Sigma_{1}$ and $\alpha I+\Sigma_{2}$ matrices, which strength diagonal properties of the $\Sigma_{1}$ and $\Sigma_{2}$ matrices. The complexity of the proposed Algorithm 2 is $O(N_{R}\log_{2}N_{R})$ instead of the original complexity $O(N_{R}^{3})$.
**Algorithm 2** Proposed Double-Step Iterative Algorithm for User Vehicle Localization1:**Input parameters:***k*, α2:**Output:**ρ(Σ(α)).3:If $x^{\left(k\right)}$ is a double-step sequence, then$\;x^{\left(\frac{2k+1}{2}\right)}=M_{1}^{-1}N_{1}x^{\left(k\right)}+M_{1}^{-1}b$, and$\;x^{\left(k+1\right)}=M_{2}^{-1}N_{2}x^{\left(\frac{2k+1}{2}\right)}+M_{2}^{-1}b$, where$\;\Sigma \in C^{N_{v}\times N_{v}},\Sigma =M_{q}-N_{q}\left(q=1,2\right)$  and Σx=b for all initial vectors $x^{\left(0\right)}\in C^{N_{u}}$.4:If k=0,1,…, then compute$\;x^{\left(k+1\right)}=\Sigma \left(\alpha \right)x^{\left(k\right)}+\xi \left(\alpha \right)b$, where$\;\Sigma \left(\alpha \right)=M_{2}^{-1}N_{2}M_{1}^{-1}N_{1}$,$\;¸\left(\alpha \right)=M_{2}^{-1}\left(I+N_{2}M_{1}^{-1}\right)$, and$\;\rho \left(\Sigma \left(\alpha \right)\right)\leq \sigma \left(\alpha \right)<1,\forall_{\alpha }>0.$5:When $\alpha \neq \alpha_{max}$,  Repeat step 4,6:When $\alpha =\alpha_{max}$,  Stop convergence, i.e., $\rho \left(\Sigma \left(\alpha \right)\right)=\rho \left(I\left(\alpha_{max}\right)\right)$.

## 4. Simulation Results and Discussion

In this section, we compare the proposed RIS-assisted CG-LHB algorithm against the conventional hybrid beamforming (G-HB) algorithm via computer simulations. Throughout the simulations, we considered the simulation parameters of Table 1. The signal-to-noise-ratio (SNR) at the receiver is defined as $\Gamma =\frac{P}{N_{s}\sigma_{z}^{2}}{\left||\Upsilon |\right|}_{F}^{2}$, where $\Upsilon =G\Theta CF_{A}F_{D}$, P=1 watt, $\sigma_{z}^{2}=-174+10\log_{10}B=-$90 dBm and the bandwidth B=251.188 MHz [28]. The azimuth angle of departure (AoD) and angle of arrival (AoA) are uniformly distributed in the (−π,π) and the elevation AoD and AoA are uniformly distributed in the interval (−π/2,π/2) [5,41]. Let the coordinates of the transmit intelligent vehicle, the RIS and the user vehicle are $\left(x_{t},y_{t}\right)=\left(0,0\right)$m, $\left(x_{R},y_{R}\right)=\left(110,20\right)$m, and $\left(x_{u},y_{u}\right)=\left(210,10\right)$m, respectively. We set the phase-shifters resolution b=2 bits and the algorithmic parameter $\tau =10^{-6}$, J=100, ξ=0.6 and α=3 [42] in the computer simulations.

For the line-of-sight channel (l=0 case), the RIS-assisted channel vector g is given using
(42)$$g=\sqrt{N_{R}}{\tilde{\beta }}_{0}\varrho_{t}\varrho_{r}a_{R}\left(\theta_{t}^{az},\nu_{t}^{el}\right),$$
where $\varrho_{t}$ and $\varrho_{r}$ are the transmit and receive antenna gains, ${\tilde{\beta }}_{0}∼\mathrm{CN}\left(0,10^{-0.1PL(d2)}\right)$ denotes the channel path gain, PL(d2) in the path loss in the RIS to user link and PL(d2) is defined as [27,43,44]
(43)$$PL\left(d2\right)=35.6+22.2\log_{10}\left(d_{2}\right)+X,$$
where $d_{2}$ denotes the distance between the RIS and user vehicle as shown in Figure 1, and $X∼N(0,\sigma_{X}^{2})$. The parameter $\sigma_{X}$ is set at $\sigma_{X}=5.8$ dB, respectively. Similarly, we also can measure the corresponding channel C and path loss $PL(d_{1})$ according to (42) and (43), and $d_{1}$ represents the distance from the transmitting vehicle to the RIS controller.

Figure 3 depicts the received signal power gain versus SNR. We see that the gain performance of the received signal power is showing the same in the high SNR case. For the low SNR case, the gain performance of the received signal power is around 0.30 dB as shown in Figure 3. Figure 4 shows the achievable spectral efficiency versus transmit power using user location information. The proposed CG-LHB algorithm provides a significant increase in the spectral efficiency for a 32×32 reflecting array compared to the traditional RIS-assisted hybrid beamforming [25,27] and with and without RIS-assisted hybrid beamforming method [3]. The achievable spectral efficiency of the proposed CG-LHB algorithm is about 1.30 bits/s/Hz, the conventional RIS-assisted hybrid beamforming method is around 1.01 bits/s/Hz and the hybrid beamforming without RIS is about 0.90 in Figure 4. For the proposed CG-LHB algorithm case, the achievable spectral efficiency performance is increased by approximately 0.301 bits/s/Hz/user at 1 watt transmitted power values. Figure 5 illustrates the achievable spectral efficiency versus $N_{R}$ by evaluating Algorithm 1. we also plot the spectral efficiency of the conventional hybrid beamforming with the G-HB algorithm [25,45]. The performance of the spectral efficiency is around 10% at $N_{R}=32$.

Figure 6 illustrates the convergence of the proposed algorithm with two gradient descent methods as in Algorithm 1 at ξ=0.6. By measurement of ∇f(Ω) in (32), the computational complexity of optimizing $F_{A}$ is dominated and it takes $O(N_{t}N_{RF}N_{s})$ multiplications. It is noted that the total algorithm needs $O(N_{t}N_{RF}N_{s}J)$ multiplications, where *J* is the number of iterations required to converge with a fixed value of parameter τ and J<100. Instead of the conventional gradient method (Algorithm 3 in [46]), which required $O(N_{t}^{2}N_{RF}^{2}J)$ multiplications, the proposed algorithm shows the computationally more efficient.

In Figure 7, we plot the error probability versus SNR, which relies on the equivalent effective channel. The effective equivalent channel is geometrically generated for the different object locations. To achieve a better error-probability performance, we applied a channel covariance splitting method and proposed a double-step iterative algorithm. In computer simulation, we evaluate $P_{e}$ as a function of SNR, where SNR leads the location information subject to the channel path loss, which progressively adds extra scatter positions to previous positions. From Figure 6, we observe that the proposed double-step algorithm provides higher accuracy than the conventional iterative algorithm [26] in the case of a multi-antenna user vehicle. Figure 8 compares the average bit-error-rate versus SNR of a RIS-assisted V2V MIMO system at $N_{s}=4$, $N_{t}=8$$N_{u}=8$, and $N_{R}=256$. We consider the proposed double-step iterative and conventional iterative algorithms. We observe that the proposed double-step algorithm significantly outperforms the conventional iterative algorithm. In Algorithm 2, we used α=3 to fix the spectral of the covariance matrices. We also observe that the average bit-error rate is improved by about 1.01 dB at the average bit-error rate of 0.0001. Figure 9 shows the channel state information (CSI) error bound in terms of location error ϵ. We used the location error parameter ϵ=3 m to validate the effect of the location error on the passive beamforming. If the number of $N_{R}$ is increased, the CSI error bound ϵ will vary as shown in Figure 9.

## 5. Conclusions

In this paper, we proposed RIS-assisted CG-LHB and a user vehicle localization algorithm. Utilizing the CG-LHB algorithm, we were able to significantly improve spectral efficiency in mmWave MIMO systems. In addition, for user vehicle localization, we considered a channel covariance splitting method and proposed a double-step iterative algorithm that reduces the effect of location error on the passive beamforming. We validated the effectiveness of the proposed algorithms using the conventional algorithms. Hence, the proposed RIS-assisted CG-LHB algorithm can be extended further to the machine learning-based beam alignment and location error minimization solution of the connected autonomous vehicles, which will be explored in future studies.

## Figures and Tables

**Figure 1 sensors-23-03713-f001:**
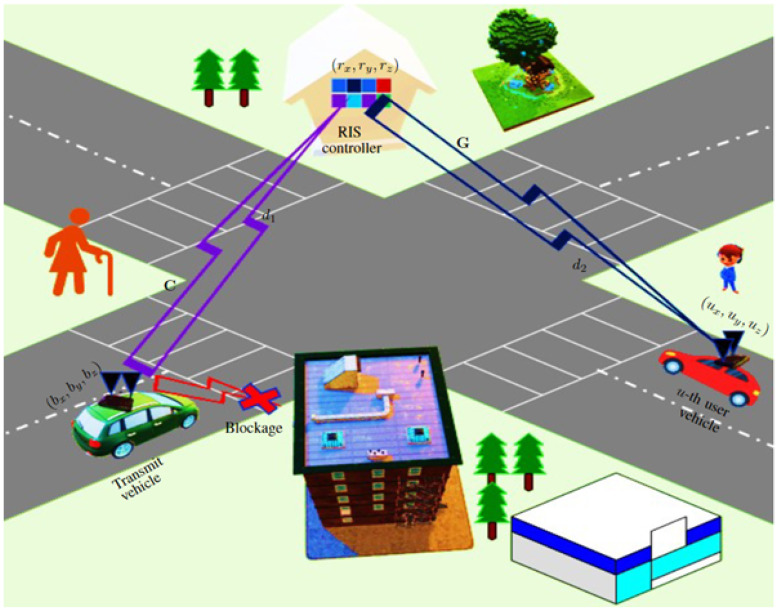
System model is illustrated by RIS-assisted V2V communications.

**Figure 2 sensors-23-03713-f002:**
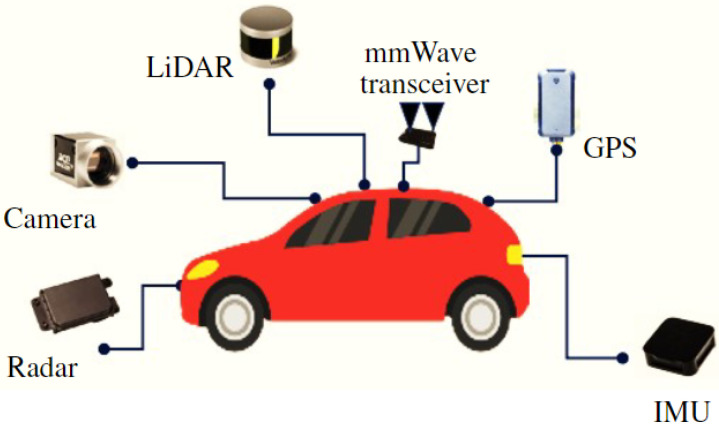
Sensors connected autonomous vehicle with WiFi-based mmWave transceiver.

**Figure 3 sensors-23-03713-f003:**
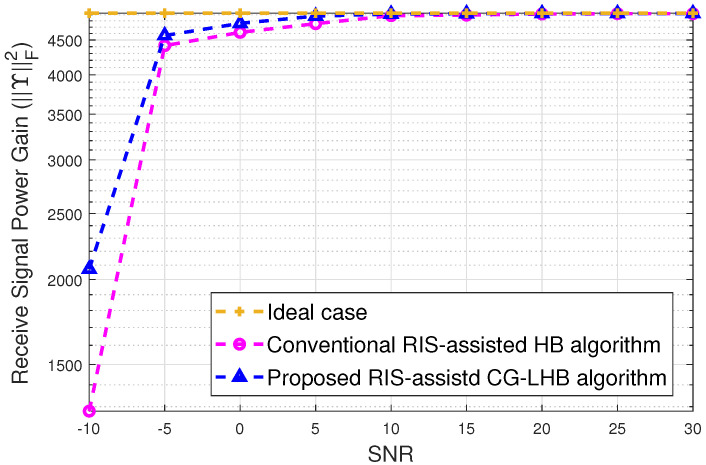
Received power gain versus SNR with the proposed CG—LHB algorithm.

**Figure 4 sensors-23-03713-f004:**
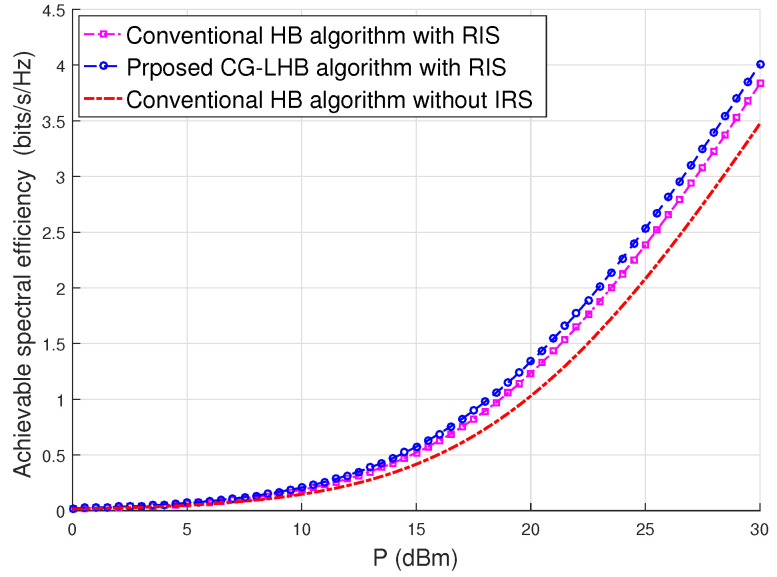
Spectral efficiency versus transmit power for the proposed RIS—assisted CG—LHB algorithm.

**Figure 5 sensors-23-03713-f005:**
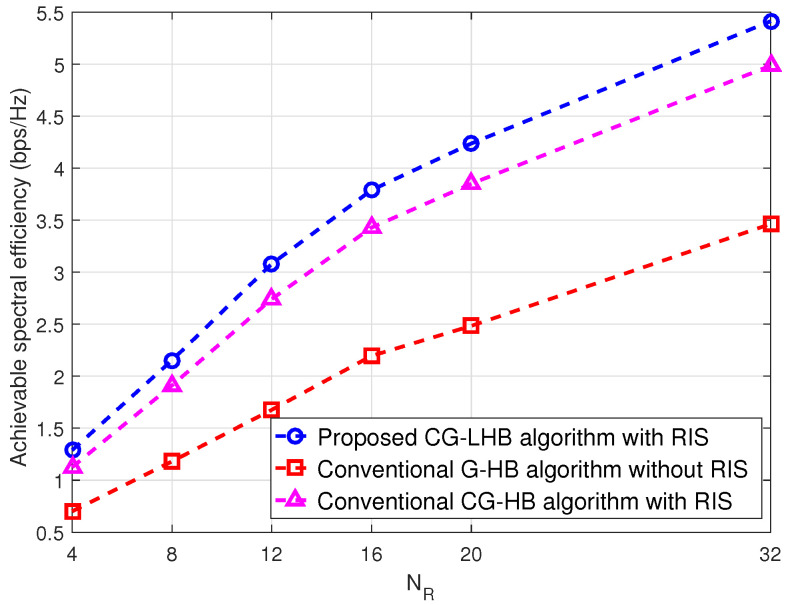
Spectral efficiency versus number of RIS elements ($N_{R}$) for the proposed RIS-assisted CG—LHB algorithm.

**Figure 6 sensors-23-03713-f006:**
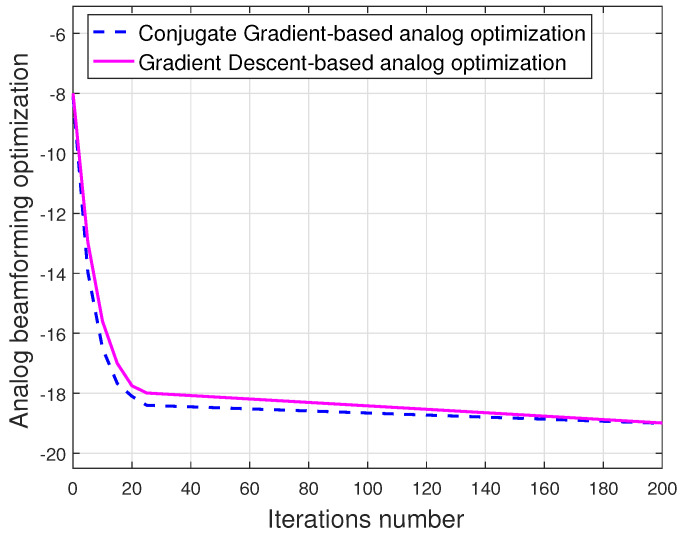
Convergence of the proposed CG algorithm—based analog beamforming optimization.

**Figure 7 sensors-23-03713-f007:**
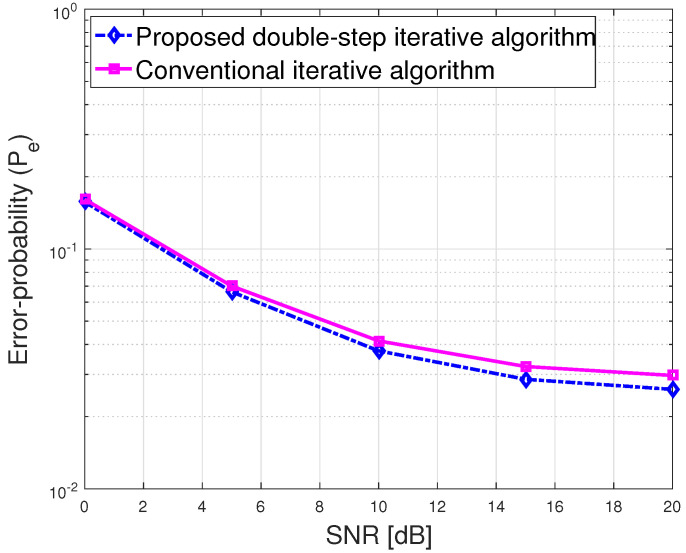
The error—probability performance of the V2V communications at 20 dB SNR.

**Figure 8 sensors-23-03713-f008:**
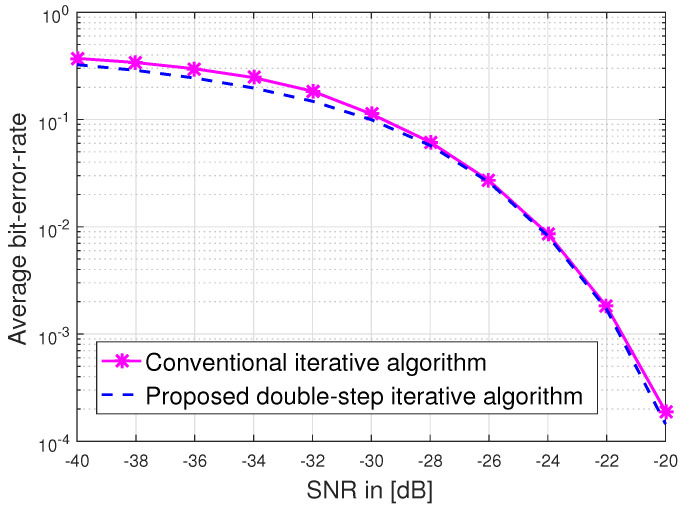
Average bit—error—rate versus SNR in a RIS system at $N_{s}=4$, $N_{t}=8$$N_{R}=256$.

**Figure 9 sensors-23-03713-f009:**
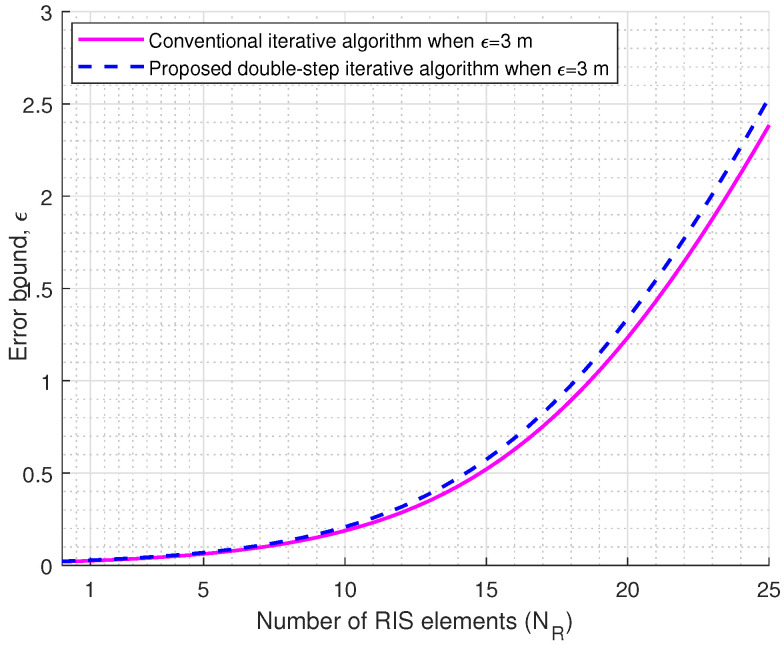
Channel state information error bound versus the number of RIS elements.

**Table 1 sensors-23-03713-t001:** Simulation Parameters.

Total number of transmit vehicle antennas	$N_{t}=8$
Total number of user vehicle antenna	$N_{u}=8$
Total number of RIS discrete elements	$N_{R}=32\left(N_{R,y}=8,N_{R,z}=4\right)$
Total number of RF chains	$N_{RF}=4$
Total number of data streams	$N_{s}=4$
The number of propagation path	1
Wavelength	λ=1 cm
Light speed	$c=3\times 10^{8}$ m/s
RIS elements distance	d=0.5 cm
Carrier frequency	$f_{c}=28$ GHz
RIS dimensions	32×32
Noise PSD	$N_{0}=-174$ dBm/Hz
Noise variance	$\sigma_{z}^{2}=N_{0}+10\log_{10}B$

## Data Availability

Not Applicable.

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
