# Peer review of "RIS-Assisted Hybrid Beamforming and Connected User Vehicle Localization for Millimeter Wave MIMO Systems"

_sensors, 2023, doi:10.3390/s23073713_

Round 1
Reviewer 1 Report (Previous Reviewer 1)
Paper may be accepted
Author Response

Reviewer 2 Report (Previous Reviewer 2)
I am frustrated by the authors' lack of understanding of what my questions are.
In Eq. (16), it is said to obtain the optimal phase shift of the RIS. Yet, the RIS phase shift never occur in (15), which is the objective function in (16). So how can the optimal phase shift be obtained from maximizing an objective funciton that is not related to it?
When a variable occurs for the first time in an equation, shouldn't you describe or define it first? I am talking about the f vector in (18) and the funciton f in (36).
Author Response
Dear Reviewer,
Thank you for your kind efforts in providing valuable suggestions. I have attached a pdf version of the response letter. Please see the attached pdf file.
Thank you for your valuable time.
Best regards,
Authors

Reviewer 3 Report (New Reviewer)
The submission is devoted to a very hot topic and is quite timely.
The review section is broad and thorough enough. Although, since, as it was said, the topic is very hot, the review can be greatly expanded (the number of publications is almost enormous). But even in this form, it covers a wide range of state-of-the-art sources.
The overall impression from the presented submission is very satisfactory. The text is well-written and well-formatted. The authors’ logic and all the derivations are clear. The obtained results are somewhat interesting.
Although, there are several issues with the submission.
Minor issues.
1. Even though the text is clear, there are some minor grammar and punctuation errors that must be fixed in the final version (if accepted).
2. Please, be careful with notations. For instance, you interchange σ and σz several times throughout the text.
3. Some problems with formatting are also present (see the power in τ (line 193 and earlier in the text)).
4. All the abbreviations must be expanded at their first appearance. For instance, what are “G-HB” and “CG-HB” in figure legends? Well, I understand, but this must be cleared out in the figure captions o in the text.
5. How do you measure the “Receive power gain”? What is this figure for? It seems to be misleading.
6. In Subsection 3.2, the authors address some “Section II-B” and “II-C”??
7. After equation (4), it is said that “ψg,l denotes the amplitude and phase for the l-th path”. It will be true if they were complex, but nothing is said about it. Otherwise, they can convey only phase information.
8. In (4), what was the reasoning for detaching azimuth and elevation for aR but not doing so for au? The same is with (7).
9. σz after equation (12) is not defined. What is it? If it was the noise variance, then why did you state that “z ∼CN (0, 1) is the Nu × 1 additive white Gaussian noise vector with zero-mean and unit variance”? This must be cleared out.
10. In line 147, the authors state that “…transmitted signal vector, which satisfies E[xxH] = I…”, but after that (in equation (2)) power multiplier is introduced. This ambiguity must be removed.
Major issues
11. What are LC and LG in equations (4) and (7)? A number of multipath rays? Then how they were estimated? And how they were set in the simulation? For a real channel model, they will be random. This will greatly affect the overall results.
12. Since some type of iterative algorithm is used (conjugate gradients), it must be thoroughly studied. It is not enough merely to state that “first fifteen iterations” is enough. This is not the way such methods are analyzed. Please, expand that part including convergence analysis.
13. The simulation and results section seems to be the weakest part of the submission. The results are not well-discussed. It must be sufficiently expanded.
14. The key question that does not leave the reader (after reading the submission) is: “How do the proposed algorithms perform against the existing ones?”
As I have several times mentioned, there is a plethora of research projects devoted to this (and very close ones) topic. The CG beamforming is not novel. Similar algorithms already exist. So what’s new? Or what is the improvement? A profound comparison of the presented results with the existing ones is absent and thus expected.
Author Response
Dear Reviewer,
Thank you for your kind efforts in providing valuable suggestions. I have attached a pdf version of the response letter. Please see the attached pdf file.
Thank you for your valuable time.
Best regards,
Authors

Round 2
Reviewer 2 Report (Previous Reviewer 2)
After three rounds of reviewing, this reviewer simply ran out of steam to continue the game. I will have to take the authors' wors and trust them have corrected the issues this reviewer raised.
Even though Eq. (18)(19) look ok now, they look totally different from Eq. (15)(16) they intend to replace. In the original (15)(16), the path gain (beta) can be solely maximized by the distances d1 and d2 with RIS plays no roles at all in the formulation. The authors don't even have the courage to admit that the original formulation is erroneous. So be it. Just don't bother me again with this kind of "research paper."
Author Response
Thank you for your kind efforts in providing valuable suggestions and accepting my answer.

Reviewer 3 Report (New Reviewer)
The authors addressed most of my concerns, sufficiently improving the submission.
Although, I still personally think that convergence analysis (even simple simulation example) must be undertaken. That will strengthen the analysis part.
Author Response
Thank you for your kind efforts in providing valuable suggestions and accepting my answer.

This manuscript is a resubmission of an earlier submission. The following is a list of the peer review reports and author responses from that submission.
Round 1
Reviewer 1 Report
1 In Abstract, it is better to add results achieved in one line and compare it with the state of art. This will help reviewer to understand the impact of your algorithm.
2. Author must highlight the simulation tool used to carry out the research work.
Author Response
Dear Reviewers,
We are pleased to re-submit the revised version Major-Revision of our Manuscript, ID: sensors-2193711, Entitled: ``RIS-Assisted Hybrid Beamforming and Connected User Vehicle Localization for Millimeter Wave MIMO Systems" in the Sensors. We appreciate and are very thankful for the constructive criticism and suggestions of the Editor and reviewers which have helped us a lot to improve the quality of our paper. We have addressed each of their concerns as outlined below. We hope that the response will serve the purpose for removing all the queries of the reviewers.
A PDF file of the response letter has been attached as a supplementary document.
Best regards,
Authors

Reviewer 2 Report
First of all, this paper is very hard to read and understand. English writing has to be improved. There seems to be some new ideas in it, but poor presentation quality prevents me further understand them. I have attached a version of the draft with places highlighted for errors or improper uses of words.
Besides wording, the paper has some math expressions that are difficult to comprehend:
1. To maximize the value in Eq. (15), I don't see why the phase in (16) matters.
2. Why do the channel samples follow the zero-mean complex Gaussian distribution?
3. The error probability defined in (36) has no tolerance margin. How could it work?
4. Proper derivation or references should be given for readers to understand the conjugate gradient-based algorithm and the matrix splitting method.
Finally, you have tp provide quantitative evidences to support that your algorithms are more computationally inexpensive.

Author Response

(The authors gave the same response as above.)

Reviewer 3 Report
Please see the attached file.

Author Response

(The authors gave the same response as above.)

Reviewer 4 Report
This paper proposes a beamforming algorithm and a user location model for a V2V RIS-assisted NLOS mmWave link. The topics of RIS communications and mmWave V2V joint communication and location are currently relevant, so at least the timeliness is good. However, the organization of the paper is extremely difficult ot follow and I am not even fully sure that I have understood the system model and proposed schemes clearly. On top of all that, even if I have read the paper correctly it appears to me that the novelty and contribution of the proposed scheme are minor. Moreover, it appears to me that the system model is quite simple and even more sophisticated systems have been studied in the literature. Detailed comments are listed below.
- It seems that the entire propsoed architecture is a single-user system. Therefore, I do not understand why the text and mathematical notations make continuous references to "the u-th user" as one would say when referring to one of the users of a multi-user system model.
-The channel model (4) only considers the LOS wave traveling form the RIS to the user, and likewise (7) for the transmitter-RIS wave. No multipath reflections are considered. This system model is in contradiction with the vast majority of mmWave sparse channel measurement and modeling literature, which has reported that the mmWave channels presents "a few" strong sparse reflections in the environment. Although several works have in fact published good contributions using a LOS-only mmWave channel model. Therefore, if they are not willing ot change the channel model, the authors should at the very least include an extended discussion of the potential influence of multipath reflections in their channel model and contribution, and write a good engineering argumentation as to why adopting a single path channel model is valid in their scheme.
- The optimal beamforming result in equation (22) appears to be well known in signal processing literature as the Zero Forcing precoder. Moreover the conjugate gradient method proposed to approximate the optimal solution using hybrid precoding should be compared with the many existing similar methods such as those in [Example1] [Example2]. In general, it seems to me that the proposed beamforming scheme is not signifincatly different from lots of prior literature. A comprehensive discussion of the state of the art and a good argumentation of the differences and advantages of the proposal in this paper should be written.
- The user location part is very difficult to read as the final goal of the location scheme is not explained until near the end of the section. I suggest reordering the paragraphs in order to explain equations (34) and (35) first and AFTER THAT explaining that there is a need to compute the matrix Sigma_u and describing how this is achieved by algorithm 2.
- The maximization in (34) is formulated analytically but the authors do not describe how this maximum can be actually evaluated, whether the computation problem is convex or NP hard or combinatorial etc.
- After understanding (34) I can go back to and finally make sense of the first half of section 3.2. Equations (29)-(30) describe the distribution of h for a given user location. I do not fully understand the role GPS coordinates in (31) and (32) in the system model though; if a GPS user location is known, there would be no need to estimate the location from the channel? I can make an educated guess that the authors meant to say that an approximate channel ĥ_e is known from approximate GPS coordinates, and the uncertain variable is Delta h? If that is the case, then this section should be significantly rewritten to convey the system model more clearly. In the opposite case, then the role of the GPS and the need of both GPS location and channel location sources should be explained.
- The improvements between proposed and conventional algoritms in simulation are very low and do not appear to justify the new proposal. The characteristics of the "conventional" system are not described. Perhaps the authors meant to say that they compare the RIS-assisted scheme vs a LOS system, and that the RIS allows to achieve the same performance in a NLOS situation? If that was the case, the section should be rewritten to state this more clearly.
- There are no simulation results for the user location contribution.
- The English language is somewhat comprehensible but many phrases are incorrectly formed. For example on page 2 "The major role of RIS can be used"-> "The major role of RIS is to be used", " the concepts of the beamforming and steering concepts" etc. In general all sections are poorly written and difficult to understand. Please consider running the manuscript by a native speaker college or conduct a thorough grammar review. Even though I tried to make the most favorable interpretation of what was not clearly readable, the contribution still appears unclear and of low innovation to me; yet I cannot fully discard that this may be an artifact of the difficulty to read the text.
[Example1] Mo, J., Alkhateeb, A., Abu-Surra, S., & Heath, R. W. (2017). Hybrid architectures with few-bit ADC receivers: Achievable rates and energy-rate tradeoffs. IEEE Transactions on Wireless Communications, 16(4), 2274–2287.
[Example2] Alkhateeb, A., El Ayach, O., Leus, G., & Heath, R. W. (2014). Channel Estimation and Hybrid Precoding for Millimeter Wave Cellular Systems. IEEE J.\ Selected Topics in Signal Processing, 8(5), 831–846.
Author Response

(The authors gave the same response as above.)

Round 2
Reviewer 2 Report
There are still issues that are not addressed properly.
1. As for the optimal phase in Eq. (16) of the original draft, the variable does not show up in Eq. (15) (and (13)(14) either) of the original draft at all; so how can it be obtained by maximizing (15)? Mind you that H = G \Theta C. The one you can control is \Theta.
2. What is the variable f in (18)? What is the function f in (36)?
3 How the modified (36) address my question about the definition of error probability? The probability, now in Eq. (33), involves the error event of the continutious random variable being equal to a certain value, which by the nature of continuous random variables has probability zero. So, how is that supposed to work out? You should either define your random variable as a discrete one, or you make a range for correct events.
Reviewer 3 Report
The authors have addressed my concerns, no further comments.
Reviewer 4 Report
I thank the authors for the time in responding to my comments. Unfortunately I believe several of my comments have not been properly addressed.
While the system model now features multiple paths for the matrices G and D, this is not taken into account in the rest of the paper. No model for the AoAs, AoDs, ZoAs and ZoDs is discussed for the indirect paths. The phase shift discussion for the RIS in Section 2.23 only considers the LOS path again. The beamforming method depends directly on H so it can absorb the change well; however the location method in section 4 does not discuss the impact of the multipath at all (in fact it does not explain how location is reverse-computed from G at all); the
While section 3 has been slightly edited, the authors still fail to explain the significant differences between their algorithm and [2,3]. Upon the reviewer comment, the authors merely cited [2,3] in the introduction without explaining how their proposal is different from prior works and without comparing it in simulation. Also only the two references provided by the reviewer were added when the comment clearly intended that the authors should have checked the literature thoroughly and found and compared further references.
Section 4 is still very unclear and difficult to understand. There is no motivation or explanation as to why, if there is one user, there are Ns different data samples to be estimated. For that matter, It still does not explain why algorithm 2, despite being called a "user location algorithm", has a probability density p(Sigma) as output. In step 3 of alg 3 values of x are iterated, but not y, why? I can't even thell if this x is the same as the x,y location or something different. I understand that (31) calculates the most likely location given a channel matrix G, and the performance of this locaiton method can change with the covariance Sigma, but I still do not understand (36) at all, I do not see a proper definition of f(Theta,Delta g) anywhere in the paper. Assuming f() is the error, then I can maybe guess that the authors are simultaneously minimizing over the RIS coefficients and finding the ML difference-of-location near the GPS location? And the RIS has the GPS location to choose the reflection coefficients? <- All of this is not explained in the text, I have only guessed and I may not even have understood it properly. These things should be explained much more clearly. Eventually, making the leap of faith that the estimation problem is properly designed, I still think that algorithm 2 and (37) appear to be obtaining the matrix Sigma itself, in order to reduce the error probability of the operation (31), but I do not understand how the vehicle position is supposed to be calculated from the value of Sigma by solving (31)? Either the authors are omitting some common knowledge that I am not familiar with (and thus should be explained to general readership for the paper to be self contained) or, more concerningly, perhaps the authors themselves have not performed this step.
The simulation section has still no demonstration of the location algorithm, only the beamforming. Figures 5-7 discuss the "communication error probability", BER, and CSI error bound; to my understanding this appears to represent the effects of algorithm 2 in the process of "finding the best value of Sigma to reduce Pe (33)" but there are no simulation results for the calculation of the vehicle-location in meteres. This increases my belief that the ecuations in Section 4 are unclearly explained and that it appears that some steps of the location procedure (31) are missing.
If its too troublesome to fix section 4, perhaps it would be better if the authors forego the "location" part of the paper entirely and focus on the beamforming method, CSI improvement and simulation results, which at least seem to be working properly?
There are still many poorly written sentences in the introduction. Specially from lines 72 to 108. Even some errors pointed out in the first review have not been fixed.